# Radiographic Outcomes of Ganz versus Modified Triple Osteotomies in Femoral Head Medialization and Coverage in Acetabular Dysplasia

**DOI:** 10.3390/jcm11071924

**Published:** 2022-03-30

**Authors:** Jui-Yo Hsu, Chia-Che Lee, Sheng-Chieh Lin, Ting-Ming Wang, Ken N. Kuo, Kuan-Wen Wu

**Affiliations:** 1Department of Orthopedic Surgery, National Taiwan University Hospital, Taipei 100, Taiwan; juiyohsu@gmail.com (J.-Y.H.); jackamades@gmail.com (C.-C.L.); dtorth76@yahoo.com.tw (T.-M.W.); kennank@aol.com (K.N.K.); 2Department of Orthopedic Surgery, Chung Shan Medical University Hospital, Taichung 402, Taiwan; phoenix33343@gmail.com; 3Cochrane Taiwan, Taipei Medical University, Taipei 110, Taiwan

**Keywords:** acetabular dysplasia, periacetabular osteotomy, modified triple innominate osteotomy, Ganz osteotomy

## Abstract

Variable techniques in periacetabular osteotomy have been formulated for the treatment of acetabular dysplasia. However, few studies have compared the radiographic outcomes between different osteotomy types. This study compared modified triple innominate (MTI) osteotomy and Ganz osteotomy with respect to radiographic outcomes. Patients receiving MTI osteotomies and Ganz osteotomies at any time between 2006 and 2018 in a tertiary medical centre were recruited. Only patients with unilateral osteotomies were recruited to eliminate potential influence from the contralateral hip following periacetabular osteotomy. Patients having hip-joint dislocation, receiving simultaneous proximal femoral osteotomy, or having fewer than 2 years of follow-up were excluded. The radiographic parameters of preoperative and postoperative anteroposterior radiographs of the pelvis were measured, and Sharp’s angle (SA), the lateral centre-edge angle (CE angle), the femoral head extrusion index (FHEI), and the centre-head distance discrepancy (CHDD) were included for comparison. Among 55 participants, 23 received MTI osteotomies and 32 received Ganz osteotomies. The mean age at which patients underwent surgery was 21.9 years in the Ganz osteotomy group and 21.1 years in the MTI group. The mean follow-up length was 2.5 years. The preoperative radiographic parameters between groups differed only slightly and nonsignificantly. Both groups exhibited significantly improved SA, LCEA, and FHEI after surgery. The Ganz osteotomy group exhibited more favourable postoperative FHEI (13.5 vs. 24.3, *p* < 0.0001), CHDD (3.7 vs. 11.5, *p* < 0.0001), Sharp angle (45.0 vs. 41.8, *p* = 0.0489) and CE angles (28.3 vs. 21.1, *p* = 0.029) compared with the MTI osteotomy group. Notably, CHDD became better and worse following Ganz and MTI osteotomies, respectively; this suggests that the femoral head is pushed laterally in modified triple osteotomy. With respect to femoral head coverage and the medialization of the femoral head, Ganz osteotomy exhibits more favourable corrections in postoperative radiographic parameters than does MTI osteotomy.

## 1. Introduction

Hip dysplasia is characterised by a shallow acetabulum with deficient coverage of the femoral head. If left untreated, it can cause early degenerative changes [1] and secondary hip osteoarthritis [2,3]. Variable techniques of periacetabular osteotomy have been used to treat developmental dysplasia of the hip (DDH). Among periacetabular redirectional osteotomies, the method of triple innominate osteotomy, first introduced by Le Coeur in 1965 [4], is one of the earliest and most representative osteotomies used in the correction of acetabular dysplasia. Several modified techniques for triple innominate osteotomy have been formulated, with variations in the osteotomy site over the ischium and the pubis [5,6,7]. In 2012, we formulated a modified approach for triple innominate osteotomy [8] with the advantage of more favourable femoral head medialisation and improved coverage when compared with the traditional approach. By contrast, Ganz periacetabular osteotomy [9] has the advantage of biomechanical stability and the capacity for large acetabular corrections even in complex clinical situations. However, few studies have directly compared outcomes between techniques, and surgical indications vary across types of osteotomies. In this study, we examined the strengths and weaknesses of modified triple innominate (MTI) osteotomy and Ganz periacetabular osteotomy by comparing correction results in radiographic parameters.

## 2. Materials and Methods

### 2.1. Study Design and Participants

Data on demographic characteristics (date of birth, sex), inpatient and outpatient records, and radiographic examination results were retrieved from the medical records of our institution. Periacetabular osteotomy was indicated if the patient had (1) symptomatic acetabular dysplasia (if they were an adolescent or young adult) with symptoms lasting ≥3 months, (2) a CE angle < 20° on anteroposterior (AP) standing radiographs, and (3) minor arthritic changes with Tönnis grade ≤ 2. A patient was excluded if they had bilateral hip dysplasia, advanced osteoarthritic changes, severe deformity of the femoral head, or less than 2 years of follow-up. We retrospectively reviewed the preoperative and postoperative X-rays from the most recent follow-up. Data on radiographic parameters were also collected. 

This study was approved by the Institutional Review Board of the National Taiwan University Hospital (202107083RIND).

### 2.2. Radiographic Measurements

We reviewed the retrospective clinical charts and preoperative and postoperative radiographs of all patients. AP standing radiographs were used to assess the extent of hip dysplasia, and radiographic parameters for acetabular morphology were collected for comparison, specifically Sharp’s angle (SA), the lateral centre-edge angle (CE angle), the femoral-head extrusion index (FHEI), and the centre-head distance discrepancy (CHDD; Figure 1). The CE angle is formed between a line drawn from the centre of the femoral head to the lateral edge of the acetabulum and a second line that is parallel to the longitudinal pelvic axis [10,11]. SA, also known as acetabular angle, is measured by a horizontal line drawn between the bilateral teardrops (inferior end projection of the acetabular fossa floor) and an additional line extending to the lateral acetabular roof [12]. FHEI is the percentage of the femoral head not covered by the acetabulum. The centre-head distance is the distance from the centreline of the body to the femoral head, and CHDD is defined as the difference in centre-head distance between the affected and normal sides [13].

### 2.3. Surgical Techniques

MTI osteotomy and Ganz osteotomy, which are periacetabular osteotomies, were performed to treat acetabular dysplasia in the participating young adults. No definite guidelines for a treatment algorithm are available for the selection of osteotomy types for DDH in skeletally mature candidates. Because Ganz osteotomy requires special instruments, modified triple osteotomy is preferred for younger individuals.

#### 2.3.1. MTI Osteotomy

In 2012, we proposed a modified approach for triple innominate osteotomy [8] with the advantage of more favourable femoral-head medialisation and coverage when compared with the traditional approach. In this study, the modified approach was implemented as follows (Figure 2A,B).

We first made an incision at the groin, extending the incision downward along the parallel line 1 cm distal to the skin crease of the groin. The superior ramus was exposed through a dissection of the area anterior to the adductor longus. The pectineus muscle and neurovascular bundle were retracted medially to expose the lateral aspect of the superior ramus. The osteotomy was made as close as possible to the acetabulum. The other osteotomy was made at the inferior ramus while the musculature of the semitendinosus, biceps femoris, and adductor magnus were retracted medially for more favourable exposure. In addition, a 1-cm autograft of bone for the osteotomy was retrieved from the inferior ramus adjacent to the acetabulum.

Another skin incision was made 1.5 cm distal to the anterior superior iliac spine (ASIS). The dissection was made between the sartorius and tensor fascia lata to expose the iliac crest; the gluteus muscles and iliacus muscles were retracted subperiosteally to expose the ileum until the sciatic notch was reached. An innominate osteotomy was made between the ASIS and the anterior inferior iliac spine. A Schanz screw was inserted into the iliac bone fragment as a joystick. The bone fragment was then rotated anteriorly and laterally with the hip abducted and externally rotated. The superior and inferior ramus were palpated to ensure the rotation and medialisation of the acetabulum. Finally, the position was reconfirmed with fluoroscopy. The previously harvested iliac bone graft was then placed at the osteotomy site and fixed with several threaded screws after adequate alignment had been achieved. The wound was then closed in layers.

#### 2.3.2. Ganz Osteotomy 

We initially began with a bikini-type Smith–Peterson incision in the inguinal area extending to the iliac wing. We retracted the tensor fasciae latae and bluntly dissected the muscle from the septum with the sartorius muscle, avoiding lateral femoral cutaneous nerve injury. We used a saw for ASIS osteotomy and to detach the sartorius. The hip was flexed and adducted to decrease the tension of the musculature, and the rectus femoris was identified and retracted laterally to expose the underlying iliocapsularis muscle. An osteotomy was made over the ASIS and reflected on the medial side. The iliocapsularis muscle was elevated from the joint capsule on the lateral side to the medial side. We opened the sheath of the psoas and retracted its muscle and tendon medially, exposing the anterior portion of the superior pubic ramus, medial to the iliopectineal eminence. We used the tip of a pair of scissors to palpate the anterior portion of the ischium at the infracotyloid groove and verified the alignment under fluoroscopy.

We placed the hip at 45° of flexion and slight adduction and inserted a 30° forked, angled bone chisel. The chisel was placed with its tip in contact with the superior portion of the infracotyloid groove of the anterior portion of the ischium, just superior to the obturator externus tendon. The position of the chisel was verified through fluoroscopy in both the anteroposterior and iliac oblique projections after we palpated the medial and lateral aspects of the ischium. The chisel should be positioned approximately 1 cm below the inferior lip of the acetabulum with its tip aimed at the ischial spine. The chisel was impacted to a depth of 15- to 20-mm through both the medial and lateral cortices of the ischium.

We carefully incised and dissected the periosteum circumferentially over the superior pubic ramus along its axis. A straight osteotomy was impacted just medial to the iliopectineal eminence. The limb was slightly abducted and extended to enable atraumatic subperiosteal dissection with a narrow elevator that was directed posteriorly towards the apex of the greater sciatic notch. Under direct vision, the iliac osteotomy was conducted with an oscillating saw and cooling irrigation aligned with the Hohmann retractor until a point approximately 1 cm above the iliopectineal line was reached.

The hip was flexed and adducted to relax the medial soft tissues, and the osteotomy was made through the medial cortex. The osteotomy was extended from the posterior end of the iliac saw-cut and passed over the iliopectineal line, through the medial quadrilateral plate, and parallel to the anterior edge of the sciatic notch as observed through iliac oblique fluoroscopy, and was subsequently directed towards the ischial spine. We used a 30° angled, long-handled chisel to connect the anterior and posterior ischial cuts to complete the osteotomy of the posteroinferomedial corner of the quadrilateral plate. We used a bone clamp as a joystick to manipulate the periacetabular bone by lifting the acetabular fragment slightly towards the ceiling, creating an initial displacement, followed by a three-step movement of lateral, distal, and internal rotation. We inserted the artificial bone graft substitutes into the osteotomy site with multiple 6.0-mm cannulated screws and fixed the ASIS with a 4.5-mm cannulated screw augmented with a transosseous suture. A 1/8” hemovac drain was inserted, and the wound was then closed by layers. (Figure 3A,B).

In most cases, we adopted a minimally invasive approach. The approach involved a mobile operative window and the utilisation of cannulated screw fixation. In doing so, we could limit the wound length to 8–10 cm.

### 2.4. Postoperative Care Programme

After index operation, the patient was transferred to the orthopaedic ward for postoperative care. In most patients, the postoperative pain ranged from visual analogue scale (VAS) 3 to 5, which can be largely controlled by use of oral analgesics and intravenous opioids as required. Ambulation with a walker was encouraged on postoperative day 1, if tolerable. A postoperative X-ray was usually taken on postoperative day 1. For MTI and Ganz osteotomies, the average hospital stays were 10−14 and 7−10 days, respectively. Weight bearing was allowed 3 and 1 months after the operation for MTI and Ganz osteotomies, respectively. When indicated, the implants were removed at 6 to 12 months after operation.

### 2.5. Statistical Analysis

Chi-square and t-tests were used to determine whether the demographic characteristics differed between the MTI and Ganz groups. Paired t tests were used to compare the preoperative and postoperative radiographic parameters. Statistical analysis was performed using SAS Version 9.4 (2014, SAS Institute, Cary, NC, USA).

## 3. Results

In total, 55 young skeletally mature adults receiving surgical correction for unilateral DDH in a tertiary medical centre in any period from 2006 to 2018 were recruited for an analysis of radiographic outcomes. Of the 55 patients, 23 received MTI osteotomies and 32 patients received Ganz osteotomies. The mean follow-up length was 30.0 months (2.5 years). In the MTI and Ganz osteotomy groups, the mean ages at which the patients received the operations were 21.9 and 21.1 years, respectively. In the modified triple osteotomy group, the mean follow-up period was significantly longer than that of the Ganz osteotomy group (47.4 vs. 16.5 months, *p* = 0.0002). Furthermore, 54.5% and 78.6% of the patients in the MTI and Ganz osteotomy groups, respectively, were classified as having Tönnis grade 1. However, both groups did not significantly differ in age, sex, or Tönnis grade. Both groups also did not differ with respect to the preoperative radiographic parameters of preoperative SA, CE angle, FHEI, and CHDD for the assessment of hip dysplasia (Table 1).

Regarding the surgical outcomes of hip dysplasia correction, both types of osteotomy showed significant improvement in SA, CE angle, and FHEI (Table 2). In the MTI osteotomy group, an average 8.1° reduction in SA (preoperative 53.1° vs. postoperative 45.0°, *p* = 0.0079), and an average 15.7° improvement in CE angle (preoperative 5.4° vs. postoperative 21.1°, *p* < 0.0001) were observed. The mean postoperative FHEI was 24.3, which demonstrated improvement when compared with preoperative FHEI (38.4, *p* < 0.0001). However, for CHDD, we observed relatively poor corrections following modified triple osteotomy, although the difference was not statistically significant (preoperative 8.8 vs. postoperative 11.5, *p* = 0.0953). 

For the Ganz osteotomy group, an average 9.2° reduction in SA (preoperative 51.0° vs. postoperative 41.8°, *p* < 0.001) and an average 19.6° improvement in CE angle (preoperative 8.7° vs. postoperative 28.3°, *p* < 0.001) were observed. The mean postoperative FHEI was 13.5, which indicated improvement when compared with the preoperative FHEI (39.2, *p* = 0.0113). Radiographic outcomes for CHDD demonstrated a difference compared with those in the modified triple osteotomy group. In the Ganz osteotomy group, slight improvement in CHDD was observed, although no statistically significant difference (preoperative 4.9 vs. postoperative 2.4, *p* = 0.0522) was observed.

Regarding postoperative outcomes, Ganz osteotomy had more favourable postoperative FHEI (13.5 vs. 24.3, *p* < 0.0001), CHDD (3.7 vs. 11.5, *p* < 0.0001), Sharp angle (45.0 vs. 41.8, *p* = 0.0489) and CE angles (28.3° vs. 21.1°, *p* = 0.0029) compared with MTI osteotomy (Table 3). Ganz osteotomy had slightly larger corrections in both SA (9.2°) and CE angle (19.6°) relative to MTI osteotomy (8.1° for SA and 15.8° for CE angle).

## 4. Discussion

Due to the scarce body of literature comparing surgical outcomes between periacetabular osteotomies, no global consensus or guidelines indicating the surgical treatment for DDH exists. Our study investigated the differences in radiographic outcomes between different osteotomy types and provided further information for treatment considerations.

Plain film evaluation is well known to play a key role in DDH surgical treatment. Feeley et al. [14] investigated the current practise of osteotomy for the surgical correction of DDH, reporting that most institutions utilised only pelvic radiographs to determine the necessity for osteotomy. Only a few studies have used advanced imaging modalities such as CT or MRI, and the advantages of such imaging techniques have not been fully established [15,16,17]. Accordingly, the present study included several key radiographic parameters that can be reliably measured without the need to compensate for pelvic tilt and rotation on an anteroposterior pelvic radiograph (i.e., without the necessity of a true lateral radiograph) as indicated in previous studies [18]. Acetabular coverage was quantified using the lateral CE angle. The normal lateral CE angle is between 26° and 42°, whereas CE angles of less than 26° usually suggest hip dysplasia [11]. Although both osteotomy types yielded substantial improvements in CE angle, the Ganz osteotomy achieved a mean postoperative CE angle of 28.3°, suggesting relatively normal acetabular coverage when compared with that of the modified triple osteotomy. 

SA is another radiographic parameter for quantifying acetabular development. Larger SAs suggest an underdeveloped or dysplastic hip with increased concentrated force that is prone to early articular cartilage degeneration and the early development of hip osteoarthritis. A normal SA ranges from 33° to 38°, and an SA of more than 42° indicates dysplasticity. According to our findings, Ganz osteotomy restored hip morphology and achieved a mean postoperative SA of 41.8°, and MTI osteotomy achieved an 8.1° improvement in SA, although the mean postoperative SA was 45.0°, suggesting a more dysplastic morphology. 

Femoral-head extrusion, which has been established to be a risk factor for the progression of osteoarthritis [19], was also measured in this study. The reference for FHEI is 17–27% [20]. An undercorrected hip with a postoperative extrusion index greater than 20% increases the individual’s risk of developing end-stage osteoarthritis and should be corrected [21]. In our study, Ganz osteotomy had a more favourable FHEI (13.5 vs. 24.3, *p* = 0.0059) than MTI osteotomy. Although both osteotomies yielded significant improvements in FHEI, the MTI osteotomy resulted in a slight undercorrection with a mean postoperative FHEI of 24.3%. By contrast, the overcoverage resulting in femoroacetabular impingement (FAI) is concerning. However, currently established values for FHEI for overcoverage in connection with prognosis are lacking. In addition, the postoperative FHEI in the Ganz osteotomy group was comparable to a previous published series [21,22,23]. We also observed no symptoms or signs of FAI in patients that received Ganz osteotomy during follow-up of up to 44.7 months.

CHDD, first introduced by Chen et al. [13], is a radiographic parameter we adopted to quantify the degree of lateral displacement of the hip joint. Lateral displacement is frequently found in DDH, both in children and skeletally mature patients [24,25,26]. Higher lateral displacement may be prone to early degeneration and be followed by hip osteoarthritis. A CHDD of less than 6% is a reliable threshold that suggests a favourable prognosis. Although Ganz osteotomy improved CHDD and achieved a mean postoperative CHDD of 2.4%, we observed the opposite—slightly increased CHDD—following MTI osteotomy, suggesting that Ganz osteotomy offers more favourable corrections than MTI osteotomy does with respect to femoral-head coverage and medialisation of the femoral head. This may be attributed to difficulties in reorienting the acetabulum when using the modified triple osteotomy technique, particularly in cases of severe dysplasticity. The attached muscular and ligamentous structure during triple innominate osteotomy may prevent mobilisation of the osteomised fragment, leading to the extrusion and lateralisation of the hip joint.

To the best of our knowledge, this is the first comparative study of modified triple osteotomy and Ganz osteotomy for the surgical correction of DDH in skeletally mature candidates. Our findings suggest that both modified triple osteotomy and Ganz osteotomy improve hip morphology with respect to radiographic parameters. Furthermore, no major complications, such as nonunion, major neurovascular injury, or perioperative infection, were observed in this study’s patients during the follow-up. Although MTI osteotomy exhibited an average improvement of 15.7° in CE angle (preoperative 5.4° vs. postoperative 21.1°, *p* = 0.0004), an average improvement of 19.6° in CE angle (preoperative 8.7° vs. postoperative 28.3°, *p* < 0.001) was observed with Ganz osteotomy. In addition, Ganz osteotomy yielded more favourable postoperative FHEI and CHDD than MTI osteotomy did. Accordingly, this suggests that Ganz osteotomy may be more suitable when the desired correction of the CE angle exceeds 20° or requires a smaller FHEI. 

Our study has some limitations. First, we did not include postsurgical functional outcomes in our analysis. In addition, we only considered the radiographic outcome of unilateral DDH. Nonetheless, in doing so, we eliminated the potential influence of the contralateral hip following periacetabular osteotomy. We also observed that the MTI osteotomy group had a significantly longer follow-up time than the Ganz osteotomy group did. However, the influence on radiographic outcome is minimal as there is little remodelling potential of the pelvis and acetabulum in our study population (i.e., skeletally mature adults).

Second, we had a relatively small sample because we excluded individuals with bilateral DDH or those who had undergone concomitant procedures. In general, however, our sample size and set of outcomes are comparable to those of previous studies on the outcomes of periacetabular osteotomies [6,27,28]. Nonetheless, future studies should evaluate long-term outcomes, such as the progression of osteoarthritis or the incidence of femoroacetabular impingement following osteotomy, in skeletally mature DDH patients.

## 5. Conclusions

Both MTI osteotomy and Ganz osteotomy improve SA, LCEA, and FHEI after operation. The femoral head may be pushed laterally in MTI osteotomy, as indicated by a better and worse postsurgical CHDD in Ganz and MTI osteotomies, respectively. With respect to the femoral head coverage and the medialisation of the femoral head, Ganz osteotomy yields more favourable corrections in postoperative radiographic parameters than does MTI osteotomy.

## Figures and Tables

**Figure 1 jcm-11-01924-f001:**
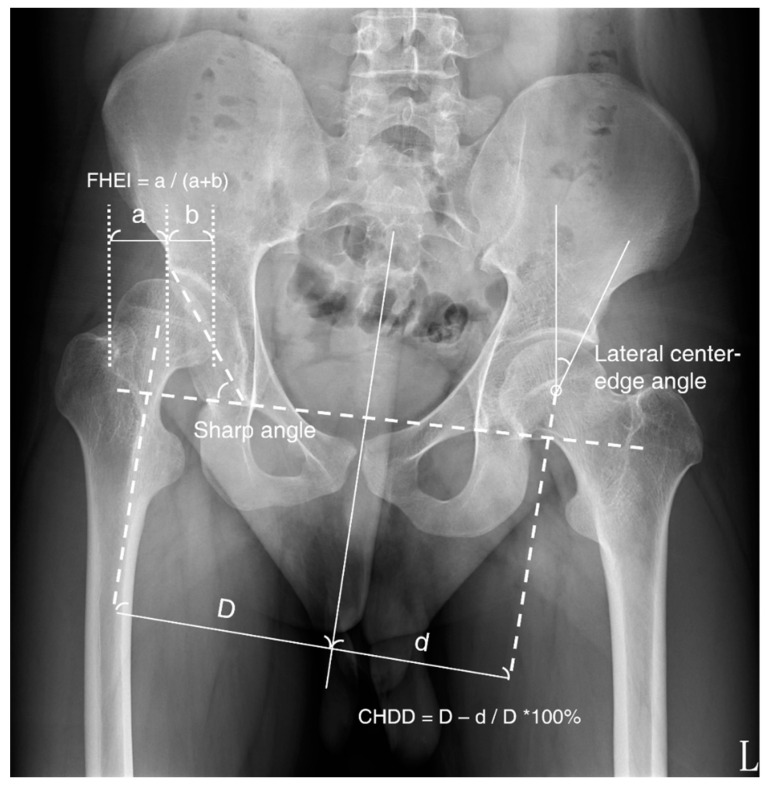
Radiographic outcome parameters on AP standing radiographs of the pelvis. FHEI: femoral-head extrusion index, CHDD: centre-head distance discrepancy.

**Figure 2 jcm-11-01924-f002:**
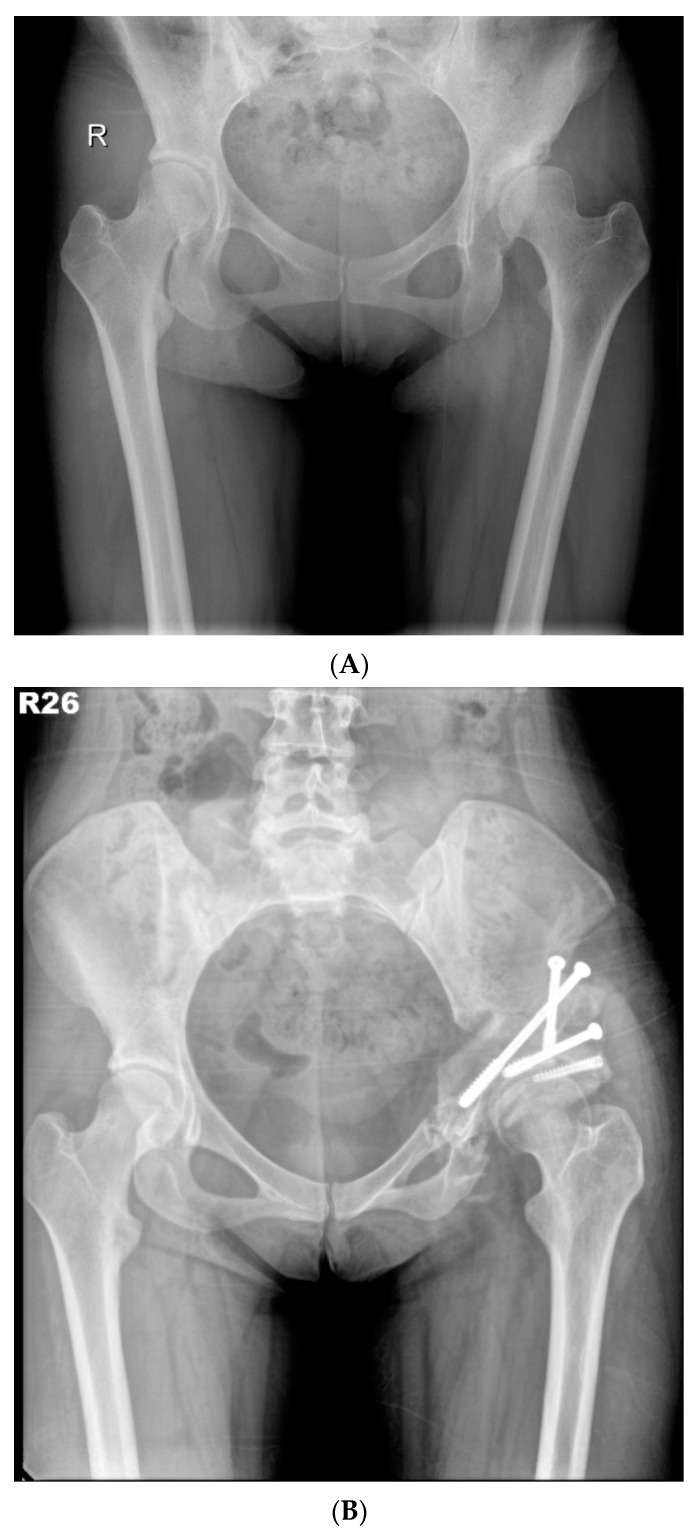
(**A**) Preoperative anteroposterior standing radiograph of a 26-year-old woman who underwent MTI osteotomy for symptomatic left hip dysplasia. (**B**) Postoperative anteroposterior standing radiograph showed good coverage of the femoral head.

**Figure 3 jcm-11-01924-f003:**
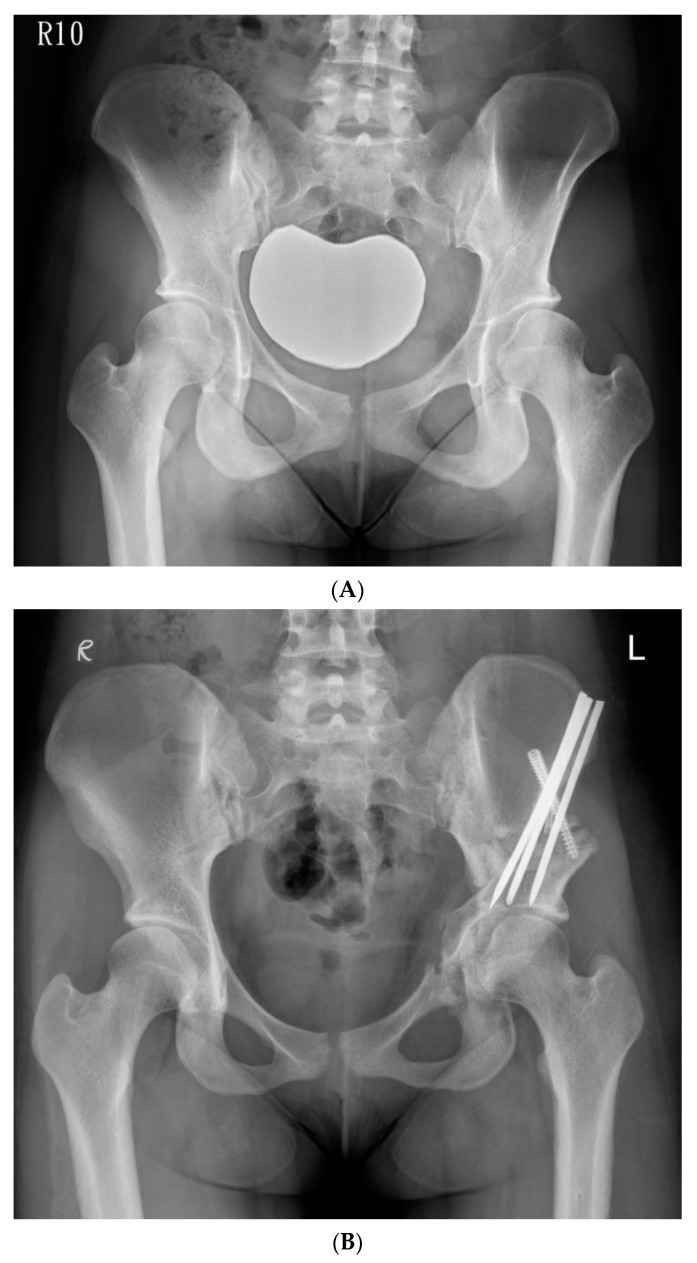
(**A**) Preoperative anteroposterior standing radiograph of a 19-year-old woman who underwent Ganz Osteotomy for symptomatic left hip dysplasia. (**B**) Postoperative anteroposterior standing radiograph showed good coverage of the femoral head.

**Table 1 jcm-11-01924-t001:** Baseline Characteristic Comparison Between Patients Receiving Modified Triple Osteotomy and Ganz Osteotomy.

	Modified Triple Osteotomy		Ganz Osteotomy		*p*-Value
Total, *n*	23		32		
Mean Age, years (SD.) ^†^	21.9	[11.1–30.3]	21.1	[12.4–29.3]	0.8320
Follow-up length, months ^†^	47.4	[24.7–80.6]	16.5	[24.1–44.7]	0.0002 *
Sex					0.4522 ^§^
Male, *n*	3	(13.04%)	8	(25.0%)	
Female, *n*	20	(86.96%)	24	(75.0%)	
Tönnis grade					0.1497
Grade 1	13	(54.5%)	24	(75.0%)	
Grade 2	10	(45.5%)	8	(25.0%)	
Pre-operative radiographic parameters					
	Mean	SD.	Mean	SD.	
Sharp angle	53.1	(3.9)	51.0	(4.9)	0.0945
CE angle	5.4	(4.7)	8.7	(7.0)	0.0549
FHEI	38.4	(9.7)	39.2	(11.9)	0.7920
CHDD	8.8	(7.1)	6.5	(4.8)	0.1573

* *p* < 0.05, indicating statistical significance. ^§^ Chi-square statistic with Yates continuity correction was applied as there is very low proportion of male in the MTI group. ^†^ The data are expressed in terms of the median and range. CE angle: centre-edge angle, FHEI: femoral-head extrusion index, CHDD: centre-head distance discrepancy.

**Table 2 jcm-11-01924-t002:** Comparison of Preoperative and Post-Operative Radiographic Parameters Using Paired T-Test in The Modified Triple Osteotomy Group and Ganz Osteotomy Group.

	Modified Triple Osteotomy				*p*-Value	Ganz Osteotomy				*p*-Value
	Pre-OP		Post-OP			Pre-OP		Post-OP		
	Mean	SD.	Mean	SD.		Mean	SD.	Mean	SD.	
Sharp angle	53.1	(3.9)	45.0	(8.0)	<0.0001 *	51.0	(4.9)	41.8	(3.5)	<0.0001 *
CE angle	5.4	(4.7)	21.1	(10.8)	<0.0001 *	8.7	(7.0)	28.3	(6.2)	<0.0001 *
FHEI	38.4	(9.7)	24.3	(3.3)	<0.0001 *	39.2	(11.9)	13.5	(6.6)	<0.0001 *
CHDD	8.8	(7.1)	11.5	(2.7)	0.0953	6.5	(4.8)	3.7	(6.4)	0.0522

* *p*-value < 0.05, with statistical significance.

**Table 3 jcm-11-01924-t003:** Comparison of correction in radiographic parameters between the Triple osteotomy group and Ganz osteotomy group.

	Triple Osteotomy		Ganz Osteotomy		
	deg	SD.	deg	SD.	*p*-Value
Post-OP FHEI	24.3	(3.3)	13.5	(6.6)	<0.0001 *
Post-OP CHDD	11.5	(2.7)	3.7	(6.4)	<0.0001 *
Post-OP Sharp angle	45.0	(8.0)	41.8	(3.5)	0.0489 *
Post-OP CE angle	21.1	(10.8)	28.3	(6.2)	0.0029 *
Mean change of Sharp angle	8.1	(8.0)	9.2	(3.5)	0.4914
Mean change of CE angle	15.8	(9.1)	19.6	(7.2)	0.0898

* *p*-value < 0.05, with statistical significance.

## Data Availability

Data is contained within the article.

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
