# Peer review of "Radiographic Outcomes of Ganz versus Modified Triple Osteotomies in Femoral Head Medialization and Coverage in Acetabular Dysplasia"

_jcm, 2022, doi:10.3390/jcm11071924_

Round 1
Reviewer 1 Report
Very good article, with adequate methods and a clear presentation of data. Language is good.
I suggest only to add Xrays of 2 patiens treated with the 2 different techniques.
Author Response
Thank you very much for the suggestion. We have included the preoperative and post-operative X-ray of each osteotomy in the revised manuscript.

Reviewer 2 Report
This study examines the radiographic outcomes of Ganz osteotomy and MTI osteotomy for acetabular dysplasia. It provides a theoretical basis for the choice of surgical procedure for acetabular dysplasia. This is a comprehensive study that advances our field but there are some concerns that need to be addressed. These are detailed below.
- For the gender distribution analysis of the Ganz group and the MTI group, the author needs to consider whether it is necessary to use chi-square test with continuity correction, because the proportion of male is very low, especially in the MTI group.
- As described in Table 1, there was a significant difference in follow-up time between the Ganz and MTI groups, would this have an impact on the assessment of radiological outcomes?
- The data for the Tönnis grade of the Ganz group in Table 1 appear to be incorrect, the sum of Grade1 and Grade2 is not consistent with the number of patients.
- Evaluation of acetabular dysplasia treatment based solely on radiographic data is inadequate, and additional clinical follow-up (eg. hip survivorship, HOOS, and UCLA Activity Score) is important to improve the reliability of the study.
Author Response
- Considering relatively low proportion of male in our study population, we have applied chi-square statistic with Yates continuity correction as suggested. There was no significant difference in terms of gender distribution between the MTI osteotomy group and Ganz osteotomy group (p-value=0.4522).
- We did observe that MTI osteotomy group had significant longer follow-up time than Ganz osteotomy group did. However, the influence on radiographic outcome is minimal as there is little remodeling potential of pelvis and acetabulum in our study population (i.e skeletally mature adults).
- Thank you for pointing out the mistake. The error has been corrected in the revised version and we have rechecked all the data in all tables to ensure the correctness.
- This study aims to compare the strengths and weaknesses of modified triple innominate (MTI) osteotomy and Ganz periacetabular osteotomy in terms of radiographic morphology. Indeed, one of the limitations is that functional outcome and hip survival was beyond the scope of this study, which has been stated in the discussion part of this manuscript. Nonetheless, our study group are currently looking into other outcome measures following periacetabular osteotomy and are looking forward to providing further information for treatment considerations in this regard.
